# Fibre-Reinforced Foamed Concretes: A Review

**DOI:** 10.3390/ma13194323

**Published:** 2020-09-28

**Authors:** Mugahed Amran, Roman Fediuk, Nikolai Vatin, Yeong Huei Lee, Gunasekaran Murali, Togay Ozbakkaloglu, Sergey Klyuev, Hisham Alabduljabber

**Affiliations:** 1Department of Civil Engineering, College of Engineering, Prince Sattam Bin Abdulaziz University, Alkharj 11942, Saudi Arabia; h.alabduljabbar@psau.edu.sa; 2Department of Civil Engineering, Faculty of Engineering and IT, Amran University, Amran 9677, Yemen; 3School of Engineering, Far Eastern Federal University, 8, Sukhanova Str., 690950 Vladivostok, Russia; roman44@yandex.ru; 4Higher School of Industrial, Civil and Road Construction, Peter the Great St. Petersburg Polytechnic University, 195251 St. Petersburg, Russia; vatin@mail.ru; 5Department of Civil and Construction Engineering, Faculty of Engineering and Science, Curtin University, Miri 98009, Sarawak, Malaysia; yhlee@civil.my; 6School of Civil Engineering, SASTRA Deemed to be University, Thanjavur 613404, India; murali_220984@yahoo.co.in; 7Ingram School of Engineering, Texas State University, San Marcos, TX 78667, USA; togay.oz@txstate.edu; 8Department of Theoretical Mechanics and Resistance of Material, Belgorod State Technological University Named after V.G. Shukhov, 308012 Belgorod, Russia; klyuyev@yandex.ru

**Keywords:** applications, foamed concrete (FC), fibres, fibre-reinforced FC, flexural strength, tensile strength, properties, shrinkage

## Abstract

Foamed concrete (FC) is a high-quality building material with densities from 300 to 1850 kg/m^3^, which can have potential use in civil engineering, both as insulation from heat and sound, and for load-bearing structures. However, due to the nature of the cement material and its high porosity, FC is very weak in withstanding tensile loads; therefore, it often cracks in a plastic state, during shrinkage while drying, and also in a solid state. This paper is the first comprehensive review of the use of man-made and natural fibres to produce fibre-reinforced foamed concrete (FRFC). For this purpose, various foaming agents, fibres and other components that can serve as a basis for FRFC are reviewed and discussed in detail. Several factors have been found to affect the mechanical properties of FRFC, namely: fresh and hardened densities, particle size distribution, percentage of pozzolanic material used and volume of chemical foam agent. It was found that the rheological properties of the FRFC mix are influenced by the properties of both fibres and foam; therefore, it is necessary to apply an additional dosage of a foam agent to enhance the adhesion and cohesion between the foam agent and the cementitious filler in comparison with materials without fibres. Various types of fibres allow the reduction of by autogenous shrinkage a factor of 1.2–1.8 and drying shrinkage by a factor of 1.3–1.8. Incorporation of fibres leads to only a slight increase in the compressive strength of foamed concrete; however, it can significantly improve the flexural strength (up to 4 times), tensile strength (up to 3 times) and impact strength (up to 6 times). At the same time, the addition of fibres leads to practically no change in the heat and sound insulation characteristics of foamed concrete and this is basically depended on the type of fibres used such as Nylon and aramid fibres. Thus, FRFC having the presented set of properties has applications in various areas of construction, both in the construction of load-bearing and enclosing structures.

## 1. Introduction

Foamed concrete (FC) has recently become widespread building material for thermal insulation and structural purposes [1,2]. FC’s growing interest caused its excellent thermal isolation performance, the ability to dispose of various waste and other important features. [3,4,5]. At its core, FC is made from a concrete mixture into which pre-prepared foam is introduced, creating a system of closed voids within the hardened composite [6,7]. FC, which is one of the varieties of cellular concrete, attracts much attention of builders worldwide [8]. Foamed concrete has nice workability and belongs to lightweight composite [7]. FC can be used for thermal and sound isolation, flame protection as well as blast viscosity; nevertheless, low mechanical and physical characteristics of FC significantly limit the scope of its application in concrete structures. [9]. Like all other concretes, it has an order of magnitude lower tensile strength compared to compressive strength. Moreover, attributable to a large amount of entrained air, the hardening mixture is subject to shrinkage to a large extent. As with other microstructure concretes, the flexural and tensile strength of FC is 15 to 35% of its compressive strength [8,10]. The utilization of different fibres in the concrete system has been found to reduce shrinkage cracks and improve mechanical properties, particularly tensile and flexural ones [11]. Recently vital energy-efficient and environmentally friendly building technologies are driving the development of green composites [10,12]. The basis for developing a wide range of these lightweight composites is pure cement combined with fine sand and discrete, evenly spaced, micro or macroscopic air cells [5,10,13]. As a result, FC provides the above advantages, reduces construction expenses and allows for sustainable designs with low weight [7,14,15,16]. Often, foamed concrete produced under construction site conditions shows terrible compressive strength, up to 9 MPa, does not allow its use as a material for supporting structures [5,8,10,13,17,18]. To increase the target compressive strength at least up to 25 MPa, engineers have applied various modern approaches that are beneficial and eco-friendly [9,17]. In the making of high strength, low water to binder ratio (w/b) and the inclusion of fly ash, silica fume, and ultrafine silica powder are recommended as a substitute to sand [19,20]. To increase FC’s mechanical characteristics, the water-binding ratio w/b is usually maximally reduced, as well as the use of finely dispersed pozzolanic raw materials as a replacement for fine aggregate [9,10,13,21,22].

Incorporation of randomly oriented fibres into foamed concrete can improve load transfer in different directions, and tensile strength increases due to the creation of an elastoplastic composite [8,23]. Enhancements of such kind can drive the creation of foamed concrete utilizing in load-bearing structures [24]. Technical papers provide a strategy for increasing the FC’s mechanical properties by adding fibres of diverse nature and various contents, embedded in the cement matrix [18,23]. Fibre-reinforced concrete has micro- and nano-fibres, which ensure the strengthening of the microstructure even in thinner cell walls [25]. Fibres existence increases FC’s flexural strength and effectively reduces shrinkage loads [26,27]. Considering that fibreglass also increases mechanical strength, this is an important factor in FC’s stability [28]. High-strength foamed concrete is obtained by introducing polypropylene (PP) fibre into the raw mix [2,18]. Many of the currently used fibres have been investigated, including PP [29,30], blend of polypropylene with glass [31], cellulose [32], kenaf [27,33], steel [26], palm oil, coconut [34], and other fibres [2,18,35]. In the previously listed studies, these fibres were introduced in an amount of 0.2–1.5% volume of the concrete mixture; however, in this article, the fibre content ranges upper limit has been extended to 5%. As an alternative to traditional reinforcement, one strategy with composite mesh and mesh combined with fibre has also been studied for lightweight foamed concrete (LWFC) [23,36,37,38]

Using synthetic and natural fibres (glass and carbon ones) in FRFC has shown excellent durability properties with reduced drying shrinkage, high modulus of elasticity and increased mechanical strength [26]. Simultaneously, it was revealed that the main target criterion of fibre-reinforced foamed concrete is density; therefore, almost all studies aims to investigate the effect of fibres and foaming agent on density. Except for some papers [29,39], the developed FRFC has a 28-day tensile and compressive strength not even more than 2.5 MPa and 50 MPa, respectively. Nevertheless, fibres are rarely utilized to enhance FC’s cured characteristics other than focusing on their effect on the performance of conventional concrete [40,41] and for the cementation of sandy soil [30].

The FC is used as potential construction materials in various civil engineering applications such as geotechnical applications, acoustic and thermal insulations, and fire resistance. But yet from being fully-used a structural concrete material due to that it has weakness at supporting tensile loads and in a plastic state it commonly cracks, in the course of drying shrinkage as well as in the hard-edged state. Therefore, the use of fibres (e.g., synthetic, metallic and natural fibres) as reinforcement is reported as an efficient material added into the fibre-reinforced FC (FRFC) matrix, resulting to bridge the cracks from extension, to yield a reduction on the shrinkage crack, to enhance load transfer and to improve the hardened properties by altering the characteristic brittle behavior to elastic–plastic behavior, particularly, the flexural and tensile properties. The fibre matrix interface and matrix densification can deliver a greater load carrying capacity of FC reliant on the toughness of the fibres. The FRFC prolongs post-crack ductility even at the case of repeating loading cycles. However, the production of FRFC efficient lightweight building materials has a major interest in the building engineering globally. 

This paper reviews the production of FC, type of foam agents, method of foaming generation, type of fibres used and factors affecting the mechanical properties of FRFC. This scientific overview also objects to provide a critical review on the properties and behaviors of FRFC as well as to synopsize the research development trends to generate comprehensive insights into the potential applications of FRFC as suitable concrete materials, for making robust FRFC composites, for modern buildings and civil engineering applications today. For the sake of clarification, this review study exhibited that there were only limited number of studies on FRFC as well as to introduce the production techniques of foaming agent and FC technology, thus, this leads the review of the literature to also include conventional FC to be able to draw a more complete picture of the current state-of-the-art.

## 2. Foam Concrete: A Preface

Romans were the first to discover agitating the mixture of small gravels, coarse sand, hot lime, water, and animal blood resulting in little gas bubbles, creating the mixture increasingly better [8,21]. Axel Erikson was the first to patent Portland cement-based FC technology in the year 1923 [9,42,43]. The characteristics, behaviour and structural applications of FC have been widely investigated by many researchers [2,6,7,14,18,44,45]. FC is reportedly proved to be superior to conventional concrete mainly due to its lower density, which helps in reducing the static construction loads, foundation volume, labour, transportation, and exploitation expenses, respectively [10,46,47,48]. In general, FC is a light cellular concrete, categorized as LWFC with density ranging between 400–1850 kg/m^3^ [2,8,18]. The mixture of the mortar’s foaming agents creates random air voids, which gives the FC good workability, lower cement and aggregate usage, and superior thermal isolation properties [46,47]. Due to its easy and robust production process, FC is known to be an economical and sustainable solution to a large number of lightweight construction materials and components [5,10,49]. Some of the FC’s widely used application is the construction of buildings, structures, filling grades, and road embankment infills. Many countries like the United Kingdom, Germany, Philippines, Turkey, and Thailand have made the use of FC in many construction applications [50,51,52]. The FC should have a 28-day compressive strength above 25 MPa if it is to be used in structural applications [9,10]. Nevertheless, utilizing FC in load-bearing structures is still not preferred because the lack of a standard code of practice and no methodology for the mix design is available [21].

### 2.1. Foaming Agents

It was proved that the foaming agents do not affect the mechanical properties much where they alter FC’s thermal properties and sorptivity [53]. Chemical foam agent governs concrete density via the amount of gas voids formed in cement pastes. Foam air-voids are known as encircled bubbles created as a result of the adding of chemical foam agents. The foam agent content has a significant influence on concrete characteristics as its states; fresh and the hardened [54,55]. The foam agent’s quality has significantly influenced the strength and toughness of the FC [56]. The bubbles’ volume varies from 6% to 35% of the total final mixture in most of FC applications [57]. 

Foam agents are inorganic and organic compounds which in pellet or powder foam. Organic agents are azodicarbonamide, hydrazocarbonamide, benzenesulfonyl hydrazide, dinitrosopentamethylene tetramine, toluenesulfonyl hydrazide, benzene sulfonyl hydrazide, azobisisobutyronitrile and barium azodicarboxylate. While inorganic agents are NaHCO_3_, NH_4_HCO_3_, (NH_4_)_2_CO_3_, and Ca(N_3_)_2_ [58]. Polyurethane foams are used as insulation spray for air-sealing buildings. Foam stabilizer is added to enhance its slurry viscosity which consists of 20% of triethanolamine 40% of polyacrylamide and 40% of hydroxypropyl methyl cellulose [59]. However, Table 1 demonstrates the various foam agents are available in E-markets worldwide.

The chemical foamer is ordinarily protein-based, hydrolysed protein, detergents, synthetic, saponin, soaps of resin, and glue resins [29]. In the mixing stage of fresh concrete, a chemical foaming agent is usually added to base-mix constituents to produce air voids via chemical reactions of foam agents [10]. Protein and synthetic foaming agents are most often used in the construction industry around the world. The protein foamers form in a more grounded and enclosed cell of air-void structure, which allows the incorporation of a more prominent volume of bubble and gives a higher steady bubble network. In contrast, the synthetic ones abdicate more prominent extension and hence lower density [29,60]. It is found that the extreme used of chemical foam volume causes a decrease in flow, density and mechanical properties of FC [55]. However, FC’s flow is considerably influenced by the time of mixing, showing that the lengthy mixing can result in the damage of the enclosed cell of air-voids by plummeting the air content [46].

Reportedly, FC’s mechanical properties are commonly dependent on the volume of foam content instead of its dependence over the proportion of water and cement [61]. In particular, FC’s mechanical properties are greatly affected by the type and volume of foaming agents, for instance, by synthetic-foam agent lower than protein-based foam agent [56,60]. It is found that the addition of air-voids in FC has less influence on elastic modulus than on hardened strength [29]. Overall, the chemical foam agent is recommended to be incorporated directly after its manufacturing in a viscous form to assure the foam’s constancy. Constancy is commonly further attained by including foam steadying fluorinated surfactant into the FC [29,55,56]. Table 2 shows the properties of FC with a different dilution of foaming [62].

### 2.2. Methods of Manufacturing Foam

Either of the two methods produces FC: (i) Pre-foaming and (ii) the mixed-foaming method (Figure 1) [5,8,73].

In the first method, fly ash (FA) is blended with water, and the foam is incorporated with the mortar or cement paste [5], whereas, when using the mixed foaming method, fly ash is then added to the slurry and the mixture is whisked into a stable mass with the required density. Methods of the foam generation have been explained below:

Pre-foaming method:-Cement, admixtures, or additives are mixed to produce the cement slurry.-About 1 kg of animal FA or high-efficiency FAs is diluted 30 to 40 times and then poured into the foam generator.-After the uniform and fine foam are emitted, it is then passed into the slurry and continued to be stirred. This is done to wholly integrate the foam and the cement slurry to make the mud cover the foam.-The FC slurry is then poured into different molds using a pouring pump to make different products.-The temperature and humidity should be maintained and kept constant for 2 to 3 days, after which the hardened FC should be cured until 7 to 10 days.-An early strength agent could be used for the rapid removal of the FC from the molds.

Mixed-foaming method:-Cement and admixtures are mixed in a mixer and stirred evenly to produce the cement slurry-A chemical FA, hydrogen peroxide, or aluminum powder, is added to the slurry and continued to be stirred for approximately 45 s.-The FC slurry is then poured into the molds of different sizes using a pouring pump to produce different products.-The temperature and humidity should be maintained and kept constant for 2 to 3 days, after which the hardened FC should be cured until 7 to 10 days.-An early strength agent could be used for the rapid removal of the FC from the molds. Moreover, Table 3 lists the comparison of foam extrusion, batch foaming and foam injection molding.

### 2.3. Production of Foamed Concrete

FC is typically composed of cement slurry or mineral binders, fine aggregate (sand or FA), and water. The production process of an FC is represented in Figure 2 [5,8].

The slurry is admixed with a FA in a concrete mixing factory [47]. After that, foam is produced when FA is admixed water with air and coming from a foam generator.

The fly ash must be capable of forming air—voids with a very significant level of constancy. It must be resilient to the chemical and physical processes of mixing, placement and curing [75]. Then the resulting foamed concrete mixes should be pumped or poured into molds or directly into structural components. Due to the foam bubbles’ thixotropic behavior, the slurry can be easily poured or pumped into the desired shapes or structural elements [47]. Depending on the ambient temperature and humidity, the slurry requires up to 24 h to solidify [76,77]. The solidification time could be reduced to 2 h with the help of steam curing at temperature up to 70 °C. After solidification, the produced foam can be released from the molds [8]. On the other hand, recent advances on FC technology are to cut the large concrete into bricks of various sizes using a suitable cutting machine with specialized steel wires. However, the concrete’s cutting should take place when the concrete is still soft [5].

## 3. Fibres Used in FC

Fibre-reinforced concrete (FRC) is one containing fibrous material that improves its structural solidity [78]. Several fibre types in concrete manufacturing more excellent resistance to impact, abrasion and destruction. A higher length of steel or synthetic fibres can completely replace reinforcement or steel in certain situations [79]. Artificial and natural fibres give various concrete properties. Besides, the FRC nature changes with a change in the concrete and fibrous materials, geometry, distribution, orientation and density. Fibre properties from previous FRFC research listed in Table 4. The appearance of some types of fibres is shown in Figure 3.

### 3.1. Natural

The benefits of natural fibres, such as good mechanical properties, low cost, density, thermal conductivity and reusability, making them an excellent potential replacement for synthetic fibres in composite materials [81]. Many natural FRC applications can be found in the construction, packaging, furniture and automotive industries [82]. Natural fibres mainly consist of hemicellulose, lignin and pectin. The composition can also change depending on the growing conditions, location and age of the plant [83].

#### 3.1.1. Abaca, Bagasse and Bamboo

Abaca is a textile banana, a perennial tropical plant. Bagasse is a dry meaty residue remaining after extracting juice from sugarcane. These fibre sources have a relatively limited growth area and weak characteristics (tensile strength 222–980 MPa, elastic modulus 6.2–27 GPa) [84,85]. Therefore, these types of natural fibre are not widely used in traditional concrete. However, due to the low density (1.3–1.5 g/cm^3^), both of these types of fibres have good prospects for use as dispersed reinforcement of foam concrete. Bamboo fibre is an innovative material obtained by regeneration of cellulose fibre produced from a bamboo stem. Despite the low density (1.3–1.5 g/cm^3^), bamboo cellulose fibres enhance fracture toughness and impact composites’ viscosity.

On the other hand, the limited regions of raw materials’ growth and the low physicomechanical characteristics of the fibres (tensile strength 222–980 MPa, elastic modulus 6.2–27 GPa) make it inappropriate for mass application [86].

#### 3.1.2. Banana, Sisal and Eucalyptus

Savastano et al. [87] compared the work on the destruction of FRFC with various fibres. FRC with banana and sisal fibres showed stable growth of fatigue cracks, while composites reinforced with eucalyptus fibre had only limited resistance to fracture and fatigue cracks. Akinyemi and Dai [88] demonstrated that banana fibres were among the best in evaluating compressive strength, tensile modulus, tensile strength, fracture toughness, toughness and energy after cracking compared to FRC with other natural fibres. Also, banana fibres have the lowest lignin content compared to other types of natural fibres, and this would require small chemical and energy resources.

#### 3.1.3. Coconut, Coir and Pineapple Leaves

Coconut fibre extracted from coconut husks is cheap and available in many tropical and subtropical countries. Coconut fibre can withstand a load of 4–6 times compared with other natural fibres [89]. The presence of coconut fibre significantly improves the flexural behavior and reduces plastic cracking of the composites. However, the results of Kochova et al. [90] showed that, despite their excellent physical properties and excellent compatibility with cement, coir fibres have low mechanical properties. This phenomenon is mainly due to the poor interface between cement and fibre. Also, fibre from pineapple leaves can be removed either by water retting and disposal or by microbe retting. It was found that the microbial retting process is more effective in obtaining fibres with a good appearance, good strength with significant chemical composition with high in cellulose and low in lignin and ash. Due to its higher cellulose content, fibre exhibits better mechanical properties than other natural fibres [91].

#### 3.1.4. Wood, Cotton and, Flax

Wood is a widely used building material worldwide. Consequently, the generation of wood waste is inevitable; it can cause serious environmental and health problems. Wood fibre is one of the most widely known natural fibres used in the textile industry. They are characterized by low density and cost, good mechanical properties, excellent dispersion, thermal properties, high corrosion resistance and heat resistance. Rongsheng Xu et al. [92] proved that wood fibre improves the aerated composite’s mechanical properties, but slightly increased thermal conductivity.

The cotton fibre obtained by grinding waste from cotton clothes contains 88–96% cellulose and this is one of the easiest types of fibre (1.3–1.6 g/cm^3^), which increases the prospects of its use in foamed concrete. Foamed concrete has a tensile strength in the range of 300–600 MPa. An elastic modulus of 6–13 GPa and an elongation limit of 6–10% make this secondary material promising of environmental protection and create building composites [93]. Furthermore, flax fibre is the oldest fibre culture globally, and its use in textiles dates back to 5000 BC [94]. By comparing with other natural fibres [95], flax gives an excellent combination of price, lightness, high strength and stiffness of the material for construction work.

#### 3.1.5. Henp and Kenaf

Hemp usually has a fibre length of 40–45 mm. It reaches up to 2 m depending on the length of the plant. In the course of fibre production, cutting is carried out into parts of 6–18 mm in size. Its chemical composition contains 78% cellulose, about 9% lignin and pectin. Since its lignin ratio is greater than that of flax, it has the shape of coarser fibres. Hemp fibre reduces the workability of the concrete mix. An amount of 2–3% and a length of natural hemp fibre of 12 mm give optimal results [96]. However, hemp the fibres comprise a considerable volume of hydroxyl groups, which increase hydrophilicity and lead to weak bond to the matrix. Surface treatment is commonly used to increase the composites performance by eliminating incompatibilities between hydrophobic matrices and hydrophilic fibres. The results exhibited that the treatment released the hydroxyl group in the cellulose and augmented its tensile characteristics due to the enhanced surface roughness.

#### 3.1.6. Jute and Palm

Of the many species of natural fibres, jute is one of the cheapest and most durable natural fibres. Jute is the second largest textile fibre in the world right after cotton. Jute fibres are mainly composed of plant materials—cellulose and lignin. Like natural fibre, jute has many convenient characteristics, such as high mechanical properties, moderate flame protection, biodegradability, renewability, processability and environmental friendliness, making it superior to other types of fibre. Traditional dosages are 0.10–1.0% of concrete volume, fibre lengths 10–25 mm. Moreover, palm fibres are among the weakest for use in composites. Tensile strength is 21–60 MPa, and the elastic modulus is 0.6 GPa, which is ten times worse than other natural fibres [91,97]. Data on ultimate stretching are not found in the open literature.

### 3.2. Artificial

#### 3.2.1. Steel, Basalt and Glass

Steel fibre is the most common among all other species. With superior durability, steel fibre is used in many construction applications. Steel fibre for concrete is a piece of the low-carbon steel wire with a diameter of 0.7–1.2 mm at a length of 25 to 60 mm [79,98]. In section, it can be round or triangular, and in configuration resemble an arc or bracket, or have a wavy shape. The fibre can improve the concrete adhesion has a rough surface. However, for foam concrete, steel fibre should be used with caution, because it significantly increases the composite’s mass. Basalt fibre is cut from a continuous basalt fibre with a diameter of 17 µm, 2.5 times superior in tensile strength to superior steel. Average fibre length: 6 to 24 mm. Recommendations for the dosage of basalt fibre: 1 kilogram per 1 cubic meter of mortar or concrete. It is necessary to introduce it into the mix with water, after stirring the fibre until it is evenly distributed in the water, it is necessary to slightly increase the mixing time (up to 5–10 min) for uniform distribution in volume. According to their chemical composition, glass fibres are divided into alkaline (containing 1–2% alkaline oxides) and alkaline (containing 10–15% alkaline oxides). For fibreglass concrete, an alkali-resistant (AR) fibre should be used. Continuous glass fibres, usually with a diameter of 10–20 µm, are obtained by drawing them from molten zirconium-containing glass melt [91].

#### 3.2.2. Asbestos

Asbestos is the collective name for a group of fine-fibre minerals from the class of silicates. In nature, these are aggregates consisting of the most delicate flexible fibres. Asbestos fibre is durable and resistant to alkalis and high heat-shielding qualities. In 1980, as a result of obtaining information about the negative impact of asbestos on human health, various non-governmental and state organizations called for the prohibition of asbestos. Since 2005, the use of asbestos in the European Union has been completely prohibited [99].

#### 3.2.3. Polyvinyl Alcohol (PVA) and Polypropylene

PVA fibres have a higher tensile strength than another synthetic one and are relatively economical in cost. Due to its low density (1.2–1.3 g/cm^3^), fibre is ideal for foamed concrete. The average fibre length is 12 mm, the diameter is 0.1 mm, the tensile strength is 1300 MPa, the elastic modulus is 25 GPa [100]. Meanwhile, polypropylene fibre, despite the low modulus of elasticity (1.5–10 GPa), has relatively high durability. Besides, polypropylene fibre has an even lower density than PVA (0.9–0.95 g/cm^3^); the fibre is ideal for foam concrete. The average fibre length is 19 mm, the diameter is 30 µm, the tensile strength is 240–760 MPa and the ultimate elongation is 15–80% [91].

#### 3.2.4. Polyethylene, Polyester and Polyethylene Terephthalate

The maximum ultimate elongation among all fibre types, both natural and artificial origin, has a polyethylene fibre (up to 100%). Yang and Li [101] reported the influence of the volume fraction of polyethylene fibre in an amount of 0.2–4% on cement composite’s destruction energy. The measured destruction energy on 2% fibre composites was about 27 kJ/m^2^, and the fracture energy dissipated by the localization of the crack was about 12 kJ/m^2^.

One of the most common thermoplastics used is polyethylene terephthalate (PET), used as bottles for drinks. General characteristics of PET: high strength, durable, resistant to damage and not biodegradable [102].

#### 3.2.5. Nylon and Aramid

Nylon fibre is one of the most common, manufactured and used in everyday life. It has good abrasion and wear resistance and good elasticity, low friction coefficient, high impact strength, fire resistance, excellent tensile properties, high water absorption capacity, good solvent resistance and high electrical insulation [103].

Aramid is a synthetic polymer, like nylon, which gives fibres of exceptional strength and heat resistance. Chopped aramid fibre plays a vital role as a reinforcing element in rubber due to its high modulus of elasticity, strength, fatigue strength, heat resistance and corrosion resistance [104]. However, the results of the use of aramid fibres in the open literature are not available.

#### 3.2.6. Acrylic and Carbon Nanofibres

Polyacrylonitrile (PAN) or acrylic fibre has one of the high modulus of elasticity (14–25 GPa), comparable with the cement matrix’s elastic modulus. PAN fibres are cheap and also have significant tensile strength. PAN fibres excellent strength characteristics are provided as a result of intermolecular forces among polymer chains, while electrostatic forces among the dipoles of neighboring –C=N groups limit bond rotation and result in a more rigid chain. It has also been found that acrylic fibres have a significant adhesion to the cementitious matrix attributable to the fibres more tremendous free surface energy than the other synthetic ones, such as nylon and polypropylene fibres [105]. These carbon nano fibres are made with a diameter of about 0.5–1.5 microns or even smaller. The performance of carbon fibres is mainly determined by the structure of graphite crystallites in their microstructures. The well-known fullerene family nano fibres are carbon nanotubes and consist of rolled graphene sheets with tubes with a high aspect ratio of more than 1000. They are various nanotubes with a tensile strength of 11–63 GPa and high modulus of elasticity from 1000–1800 GPa. Table 5 lists the hardened and physical properties of some fibres used in FC. 

## 4. Factors Affecting the Properties of FRFC

Several factors affect the mechanical properties of FRFC, namely fresh and harden densities, aggregate grading, pozzolanic effect, foams used in concrete etc. The following contents describe these factors. Table 6 illustrates the properties of different FRFC composites and mortars.

### 4.1. Fresh and Harden Densities

It can be found that finer aggregate size may stabilize the density of FC for quarry dust as sand replacements, while, charcoal as a sand replacement, exhibited high discrepancies on its fresh and harden densities. However, the finer aggregate may increase the fresh concrete density. The addition of fibres may reduce the fresh concrete density as compared to pozzolans [116,117]. On the other hand, additives like strength enhancement chemical may burst the foams observed in [118]. Therefore, the chemical reactions between foams and additives must be identified before applying to the FC mix design. The compressive strength and density ratio are mostly curvilinear for hardening concrete [119], and it can be proven by [29] of its exponential relationship. The increment became larger with silica fume addition. Porosity is inverse with density [120], where the inserted foams are greatly affected by the concrete harden density.

Therefore, the fresh density is highly depends on the individual density in fresh FRFC, namely aggregate, fibres, cement and foams. Heavier materials may cause an increase in FRFC, but also enhance the concrete strength [118,119]. Depending on the targeted density and strength, the fresh density can be manipulated with the individual densities. For harden concrete, the density is much relying on the mix stability and the pores where chemical reactions of additives may burst the bubbles at fresh stage and making more dense structures. Hence, it is recommended to use stable foams to achieving a desired harden density.

### 4.2. Diameter, Length and Content of Fibres

Three fibre factors affect the fibre reinforced concrete properties, namely fibre kind, fibre content and fibre mix-ability [133], respectively according to their compressive stress analysis. Referring to Table 6, the fibres content should be controlled within 5 % to achieve optimum concrete behaviour where excessive fibres may reduce the concrete strength. The high content of paper fibres of 20% was found in [122], where the compressive strengths were relatively low, not more than 2 MPa. Steel fibres found enhancing concrete properties with also increasing concrete density gradually. However, polypropylene is useful to be applied in FRFC, balancing the density and concrete strength. Generally, fibre improved the properties of FRFC, especially in concrete tensile or flexural strength. There are available lengths in the market for synthetic fibres, and their properties are known, whereas natural fibres are needed to be processed before applying into the concrete. Polypropylene lengths of 12 and 19 mm were investigated with high frequency from previously conducted researches. The smaller diameter is found enhancing concrete properties, as shown in Table 3. Hybrid fibres in concrete were found to increase the concrete strengths [23] gradually. From previous aerated concrete research [134], polypropylene was found sensitive to fire, oxygen and sunlight. Also, the elastic modulus and bonding with the concrete matrix are not as good as steel fibres. However, compared to plain foamed concrete, polypropylene was able to enhance concrete strength and decrease drying shrinkage [29]. 

From previous investigations, it is recommended to have a low fibre content to achieve optimum properties where high fibre content was not benefit in its properties enhancement. The geometries of natural fibres are discrepancy while synthetic fibres may have standard geometries. Therefore, using synthetic fibres may have low strength fluctuation over same proportion of the mix. Also, hybrid fibres in FRFC may enhance the concrete strength, was found in recent studies. 

### 4.3. Effect of Aggregate Grading

As claimed by Lim et al. [135], finer of the aggregate size may increase the strength of FC (Table 7).

However, it is not practical to sieve the aggregate in massive production. As the particle size grading decreased, from passing through 2.36 mm sieve to 0.60 mm and 0.10 mm sieve, the strength has an increasing trend [118]. In a massive casting, as mentioned in the slab casting research [144], it was found that the non-sieved sand may have lower strength as compared to those sieved samples. With finer aggregate, the workability also increases and reduces the use of compaction that potentially bursts the bubble inside concrete paste. Depicting the impractical of sieving the aggregate for cast insitu FRFC, it is suggested to apply in precast FRFC where the quality can be controlled through factory manufacturing.

### 4.4. Effect of Pozzolanic

Silica fume has been found to benefit concrete properties [145], especially compressive strength and durability. It was added to FRFC and was found to have a 25% increment, at most, in compressive strength [29]. Research also discovered that 70–75% strength was developed for FC without silica fume and 85–90% with silica fume (SF), corresponding to their 28-day concrete characteristic strength [29]. Moreover, silica fume also used to stabilize the foams in the concrete matrix [146]. FA, silica fume and metakaolin were investigated as a pozzolan for FRFC [120,147].

With the same quantity of addition, fly ash and metakaolin specimens have higher compressive strength than silica fume [120,147]. Other than silica fume, nano-silica also have been investigated [148]. The 4% nano-silica was found to have more excellent mechanical properties, durability and microstructure than plain FC and 15% silica fume [148]. Silica fume was effectively enhanced the flexural, compressive and tensile strength and durability with a combination of waste marble material [149]. Moreover, in order to enhance LWFC properties, apart from traditional materials, such as FA or SF [10], the recycled components like the slag of various types [150,151] or glass [23], have also been included in the LWFC concrete matrix, which introduced better solid waste management through reuse of the manufacturing by-products towards higher concrete performance. With the similar properties of LWFC and FRFC, it can be concluded that the pozzolans may increase the concrete strength and its durability, exhibited the same with normal concrete with pozzolans. 

## 5. Properties of FRFC

FC is obtained from mixing base mix (normally mortar) with preformed foam (diluted foam agent with high pressure). The material properties of FC may have a significant effect on the structural performance of the LWFC structures. The strength behaviors have been summarized, which give a significant and positive response to their structural behaviour. FC consists of cement as binders, sand as aggregates, water, and foams. As a consideration of economic and performance enhancements, many researchers were introducing additives or replacements to the FC, such as fly ash [19,29,152,153], silica fume [29,153], superplasticizer [153], fibres, and others. There is no specific method to determine the mixing proportions. However, Kearley [19] proposed the calculation of mixing the proportions by the target density method, and other researchers have practiced this.

### 5.1. Fresh State Characteristics

Fibre-reinforced foam concrete mixture is poured into molds without mechanical compaction, so it must have self-sealing rheology. For optimization various performances such as consistency, rheology and workability, segregation and bleeding should be considered [8,154]. These parameters mainly depend on the water/cement ratio, supplementary cementitious materials, aggregates, superplasticizers, foam agents, and the type and concentration of fibre. Fresh properties may affect hardened mechanical properties. As an instant, a self-compacting characteristic must be obtained in a fresh state to maintain its workability, where foams bursting may occur during concrete compaction and needs to avoid this event in concrete casting. The consistency and rheology should be attained as the mixed slurry can flow and hold the bubbles without segregation. The accepted workability should fulfil spread-ability between 40 to 60% of the 20s for a self-compacted mix [8,67] using an inverted slump test setup. There were several methods to obtain a mix’s stability based on the difference between the achieved and desired plastic density, which is not meant to exceed 2 to 7% [8,155,156]. Moreover, the workability of the base mix for FC can be accepted when achieving a spread of 85 to 125 mm for the mortar mix and 115 to 140 mm for the mix with fly ash [157,158]. 

#### 5.1.1. The Rheology and Consistency

The first measured indicators of fresh FRFC, are usually measured using a slump flow test or a rotational viscometer (shear stress measurement). The flow time should be within 20 s so that a sufficient amount of the mixture is placed in the mold and self-sealing without any external devices [8,55]. Reportedly, [159,160], various factors affect the mixture’s consistency and rheology; which are mainly associated with the components and technology of the mixture. For this, the mixture’s components must be precisely calculated to improve the rheology and consistency of FRFCC, attain self-sealing behaviors, and enhance the adhesion and cohesion between the foaming agent and the cementitious filler. One of the main features influencing the rheology and texture of fresh FRFCC is the mixture’s water content. Another main factor is the fibre density in the mixture. For instance, the adding of light fibre negatively influences the constancy of the mixture. Besides, the rheology of FRFC worsens with excess foam due to the higher air volume, while the incorporation of SP improved the flow rate. Several studies [161,162,163] of the “Portland cement—silica-containing additive-complex modifiers” system can significantly reduce shear stress and create easily compacted fibre-reinforced foamed concrete mixtures.

#### 5.1.2. Workability

FC’s workability demonstrates excellent characteristics caused by air bubbles in the mix as a result of the incorporation of a stable foam agent. The flowability test usually carried out using the slump method for ordinary concrete [164], is not appropriate for low-density fresh FRFC. The FRFC workability is evaluated visually to achieve the suitable viscosity of the mixture. When designing the FRFC composition, it needs to take into account the shape of the fine aggregates. The aggregates rounded shape reduces the likelihood of ordering of the fine aggregates and thereby increases the spreadability [79]. High values of slump lead to enhance in the average density of FC and the volume of materials used in production. The introduction of fibre somewhat reduces workability. The fibres large specific surface absorbs more of the cementitious mortar around the fibres and, consequently, increases the viscosity of concrete, which contributes to a slight reduction in the values of the spreadability. However, with proper FRFC mix design, even a specific improvement in workability can be achieved by introducing the optimum fibre amount. Fibre as an inert material (not chemically interacting with the foaming agent and water in the foam system) will “push apart” the foam films with the formation of peculiar channels in their thickness [165]. This will provide an increment in the workability of the mixes due to the free movement of the dispersed phase over the volume of the aqueous dispersion medium, including between the bubbles.

#### 5.1.3. Segregation and Bleeding

The principle of the segregation and bleeding (water separation) test is that after sampling, the concrete mixture is left for 15 min and the bleeding is monitored [98]. After that, the upper part of the sample is poured onto a sieve with a mesh size of 5 mm. After two minutes, the weight of the material under the sieve is fixed. The stratification coefficient is calculated as the mixture ratio above the sieve to the amount of mixture under the sieve. It was suggested that the ratio of w/c be reduced since an undue amount of water initiates the segregation of foam during concreting, which influences the machinability [166]. Since, in principle, water separation decreases with decreasing water-cement ratio, superplasticizers, if used to reduce w/c, do not cause separation or water separation. The use of a clogging micro filler (for example, finely ground limestone or quartz sand) helps to reduce the separation and water separation of self-compacting concrete mixtures. According to the results of [167], this is suggested that the micro filler creates an additional “stability framework, which increases the resistance to segregation of the mixture.

### 5.2. Hardened Properties

An assumption has been made that the FC had achieved fresh properties requirements before investigating its mechanical properties in the hardened stage. Figure 4 showed the air bubbles that entrapped in the concrete in the hardening stage.

Due to the fibres and the bubbles, the properties of FRFC may be different from those normal fibre-reinforced concrete. The following sections discuss the strengths of FRFC at the desired concrete curing age. Previous research has shown that the inclusion of fibres has more than one type or more than one size, recognized as hybrid fibres. 

#### 5.2.1. Compressive Strength

The mechanical properties of fibre-reinforced foamed concrete are derived from the concrete matrix and its microstructure. The compressive strength decreases exponentially when concrete’s density decreases, relative to normal concrete [8]. FC’s compressive strength commonly decreases gradually as the entrapped bubbles increased; however, its strength may be enhanced by adding fibres into the concrete matrix. Massive research [29,31,169] has shown that various fibres can enhance FC properties [6,7,8]. Also, it has been proven that these fibres can enhance its hardened properties (including fresh, physical properties, toughness and elasticity) with experimental evidence [8,23,38,170,171,172] (Table 7). To alter the brittle behaviour of FC, Amran et al. [8] added fibres to make it ductile. Moreover, polypropylene also helped to reduce the crack width with fly ash [173]. In further analysis, a combination of carbon and polypropylene fibers in FC was found to enhance bending stiffness compared to individual fiber mixtures [170]. It is also proven that the increment of flexural strength with polymer fiber of 2–5% substitution while compressive strength is maintained as the same value [23]. Previous studies of [27,174] had found polypropylene fibers are suitable and mix well with the cement-based foam matrix, which can further justified by other findings [175,176]. A complex analysis of compressive strength is required for lightweight concrete. The entrapped air to cement ratio needed to be quantified and rarely applied to normal weight concrete analysis [177]. It is also revealed that the overall strengths were mainly influenced by the factor of combined water/cement and air/cement ratio [178]. For more specified quantification, with referred to control specimen, it was found that 1% carbon fibers improved FC compressive strength by 36% while incrementing of 23% with 1% carbon fibers combined with 0.5% PP fibers [35]. The addition of 2.0 and 4.0% of steel fibres improved the strength of FC by 9% and 12%, respectively [179]. 

Traditionally, fibres inclusion leads to toughness and durability improvements, while, recently the hybrid fibres in FC may enhance its strength properties to the structural application [180]. The toughness of concrete in the post-crack stage may increase with steel wire fibres. Meanwhile, the initial crack strength may improve with mill-cut steel fibres with a better bond with concrete [181]. Furthermore, the inclusion of hybrid fibres in FC showed good workability [182] and recorded a great value of toughness index. The concrete mechanical properties for monofilament and staple fibres of hybrid polypropylene was investigated [183] and found an improvement from single fibre application [184] where staple fibres can reduce initial cracks with its good fineness and dispersion [31].

It is found that the compressive strength of FRFC was attained up to 50 MPa, with only 65% of the normal concrete density [29]. However, these may not represent the behaviour of FRFC and further empirical models needed for strength prediction with different fibres. 

For natural fibres of kenaf, a 0.45% volume fraction for reinforced FC with a density of 1250 kg/m^3^ exhibited an increment of compressive strength compared to plain FC [33] and a similar trend was found for 0.2% and 0.4% coir fibres in FC at density range of 800–1250 kg/m^3^ [34]. Paper [185] also discovered that the compressive strength was improved by including 0.75% of sisal fibre and justified by [186] for 1200 kg/m^3^ sisal FRFC. An improvement of the compressive strength of henequen FRFC was found with densities of 800 kg/m^3^ [187] and 900 kg/m^3^ [188].

The compressive strength may increase as much as 2.5 times of plain FC, and achieve structural use. The structural uses FRFC were having their density from 1800 to 2250 kg/m^3^. The heavier steel fibres were found to be the main factor in their gradually increase density and enhanced the strengths of FC, unlike kenaf, that is showed no much contribution in improving FC strengths [189]. Another study reported that the addition of 3% PP fibre on the FC of 1800 kg/m^3^ increased concrete compressive strength by 18%, while the 3% kenaf fibre instigated a 22 % decrease in compressive strength [31]. It can be achieved with similar strength by hybrid fibres of carbon and polypropylene with a density of 1800 kg/m^3^, which was more efficient [189]. 

#### 5.2.2. Modulus of Elasticity

A previous study of [190] showed fibres ability to solve the composites brittle properties, saving a life during building failure, and justified by other research of fiber reinforced lightweight concrete [72]. Moreover, it is proven that the incorporation of 0.56% of polypropylene by volume in LWFC increased 90 % of indirect tensile strength and 20% of rupture modulus [18]. Improvements of split tensile strength and shrinkage resistance have been found in [72] by adding steel fibres to FC. Specifically, FC with 0.2% and 0.4% steel fibres enhanced 9% and 12%, respectively for elastic modulus [183]. 

The addition of 3% polypropylene fibre on the FC of 1800 kg/m^3^ increased elastic modulus by 40%, while the 3% kenaf fibre instigated a 29% decrease [31]. For 0.2% and 0.4% coir fibres addition, FC densities of 800–1250 kg/m^3^ also improved in concrete elastic modulus. [34]. Furthermore, Liu et al. [185] found the improvement of shrinkage behaviour and elastic modulus was discovered for FC with 0.75% of sisal fiber. Also, FC with henequen fibres of densities of 800 kg/m^3^ [187] and 900 kg/m^3^ [188] had a research record of increment trend in concrete elastic modulus.

With referring to ACI318-05 [191], there is no formula representing the elastic modulus for density 1120 to 1440 kg/m^3^ and the existing equation is overestimated 16 to 104% of collected data [134]. There are limit published data for elastic modulus and fracture tests. Therefore, the adaption of prediction models should be through previous FC data and further verified by experiments. As shown in Table 8, the variation range is high with scatter data and unable to justify which prediction model can represent the elastic modulus of FRFC, as there is no data for elastic modulus from previous experiments. 

#### 5.2.3. Splitting Tensile, Flexural and Fracture Strength

The tensile strength of FRFC has been improved, as compared to those plain FC mix. The American Concrete Institute [193] suggested adding fibres to benefit the splitting tensile strength at an early age. Most of the FRFC can achieve the minimum requirement of 2 MPa of ASTM C330 [194] for structural concrete. From previous research [29], the equation has been developed for FC with and without polypropylene fibres, which formed with their compressive strength. The flexural strength was examined for FC with fibres, namely polyolefin [72], polypropylene [29,165] and waste tyre steel fibres [57], which added into the base mixture before foams are added. These fibres only marginally increased the flexural strength of FC. Fracture energy is the energy dissipated supplied to a growing crack tip and balanced by the energy dissipated from new surfaces formation. FC beams fracture energy was identified with three-point flexural tests on notched beam samples [195,196]. The fracture strength increased with the increments of compressive strength and density. 

Fibre addition of 0.4 to 0.6% in FC was characterised in 10–17 MPa strength class for steel fibre, glass fibres, polypropylene fibres [197]. It is indicated the high destruction energy for Bazant and Committee Euro-International du Beton models, but for the Hillerborg model, it is increased in fracture energy with fibres increment. Therefore, destruction energy will increase with increment of fibre portion and suit to all types of fibres. 

FC’s flexural strength was recorded as 4.96 MPa with 90 curing days for the 0.25% polypropylene fibres inclusion and found to be increased with curing age [198]. Similarly, it is revealed that the strength enhancement of 20–50% with polypropylene fiber and silica fume [18]. With 1% carbon fibres combined with 1.5% of carbon fibres alone and 0.5% polypropylene, the splitting and flexural strengths were increased 48 and 116%, and 44 and 116% respectively, compared with plain FC [35]. 

The tensile strength has been significantly improved for FC with density variety between 1200 and 1800 kg/m^3^, silica fume of 5–15%, and steel fiber addition range of 0.25–0.5% [199]. Similarly, enhancement was exhibited for the FC density of 1000 kg/m^3^ with steel fiber of 0.25% and 0.4% [200]. For tensile strength and flexural strength, improvements of 32% and 42%, also 34% and 26% respectively for 0.2% and 0.4% steel fibre addition [179]. For 800–1250 kg/m^3^ FC density range, 0.2% and 0.4% of coir fibres [34] and 0.75% sisal fibres [185,186] were found effectively improved tensile and flexural strengths. Also, for henequen fibres, improvements of the tensile and flexural strengths were discovered of FC with densities of 800 kg/m^3^ [187] and 900 kg/m^3^ [188]. 

The increments of 84.7% and 558% in tensile strength for the addition of 3.3% of polyvinyl alcohol fibers compared to plain FC for the density of 1000 kg/m^3^ [201] and fibres altered FC material stress-strain behaviour. With the studied range of 0.25 to 0.5% of polypropylene and 1400 kg/m^3^ FC density, it was found that optimum performance can be found from 0.5% where flexural strength was 58% higher than plain FC [21]. With the same fibres of polypropylene and different concrete density 600–1400 kg/m^3^, flexural strength was improved optimum with volume fractions of up to 0.4% [202] and similar results were obtained for 0.5 to 3% of polypropylene with a density range of 1000–1900 kg/m^3^ (Figure 5) [18].

#### 5.2.4. Water Absorption, Drying Shrinkage and Time-Dependency Properties 

The effect of drying shrinkage is significant with the absence of aggregate with low concrete density than conventional concrete [27,29]. It may benefit from fibres added to the concrete matrix by increasing its tensile strength [23,175]. Figure 6b shows that the level of reduction is negligible for the 0.7% fiber content, about 35% for the 2.0% fiber content, and around 45% for the 5.0% fiber content [23].

Drying shrinkage was reduced at 28-days with short polymer fibres, as shown in Figure 6b, where a 35% reduction for 2.0% fibers and 45% for 5.0% fibre content. For polypropylene fibre inclusion, the splitting strength was improved equal to ½ time of the identical reference specimen and exhibited drying shrinkage of (900–1300 kg/m^3^) × 10^6^ and creep of (434–734) × 10^−6^ at 150 days [2].

The hydrophobic characteristic is also found in polypropylene fibres; contrary, natural fibres like kenaf may absorb water due to surface morphology [189], which may consider in the casting process. The time-dependency properties of FC include the creep strain and drying shrinkage. Drying shrinkage occurs in the first 20 days of casting time, which significantly affects the concrete strength. Previous research showed that fibres reduced the drying shrinkage of FRFC [32,123,124]. Shrinkage may induce premature cracking and thus weaken the concrete strength. FC may have 78% higher shrinkage deformations and a 50% higher creep than geopolymer concrete [203]. FRFC may achieve 50% of the entire creep strain in the first month of loading application [80]. Creep strain and drying shrinkage should have a further investigation for a better prediction model of FRFC.

#### 5.2.5. Thermal Conductivity

The thermal performance of porous materials has been investigated since the last decade by [204,205,206,207]. The effect of pozzolan, density, foamer content, aggregate mineralogy and microstructural factors on conductivity are the scope of [208]. However, the entrapped air in concrete is the major reduction in thermal conductivity related to concrete density. FC as building materials, also known as closed-cell structures, having better thermal insulation, owning 0.1 to 0.66 W/mK for a density range of 300 to 1600 kg/m^3^ [209]. It is clear from Figure 7, the thermal conductivity was reduced in a wide range of density for cellular cementitious composite [206,210,211].

A 35% thermal reduction was observed for FC density from 800 to 600 kg/m^3^ [212]. Pozzolan materials have also been found to reduce the concrete thermal conductivity reported in [204,213] for fly ash, silica fume, metakaolin, slag and bagasse ash and in [147] for fly ash, silica fume, and slag. 

Previous research indicated that relatively high thermal conductivity reduction with polypropylene, as compared to AR-glass, kenaf, steel and oil palm fibres [36] and steel fibres would not likely increase the concrete thermal conductivity [214]. Other findings also showed the polypropylene fibres reduced concrete thermal conductivity [132]. Fibre content should be controlled at 1.0% or less for lower thermal conductivity of glass fibre FC [215]. ACI setup a prediction equation for thermal conductivity, with considering lightweight concrete density [216]. The precision of the prediction equation was yet to be confirmed for FRFC. Among oil palm, steel, AR-glass, kenaf and polypropylene was less effective in thermal insulation while polypropylene was better at a higher percentage of fibres and decreased thermal efficiency [26]. Polypropylene with 0.2% effectively reduced thermal conductivity by 7%, compared to plain FC [147]. If strength and insulation are required, it is suggested to use pozzolans like fly ash to meet adequate concrete strength and thermal behaviour [147]. A 6% reduction of concrete thermal conductivity was discovered for cement-based foam with silica fume, fly ash and metakaolin [217] while the thermal conductivity of others FRFC without pozzolan yet to be effectively determined. 

### 5.3. Summary of Properties of FRFC

In fresh state, the governed properties are focused on workability which may cause bleeding and segregation. The hydrophilic and hydrophobic characteristics of the materials in concrete matrix are associated with its workability. Hydrophobicity may cause segregation as the materials are not mixing well in the matrix, while hydrophilicity may affect its workability. The balance should be applied between hydrophilic and hydrophobic, with sufficient water-cement ratio to obtain its workability. Bleeding and segregation are the common phenomena where excessive water has been added into the mix. Workability is important in FRFC as compaction is unlikely to be applied onto concrete which may burst the foams in the vibration process. 

Meanwhile, during harden state; the mechanical and functional properties are the main focused area. With different applications, a wide range of properties can be produced for FRFC. The required specifications must be achieved for certain use, for instance, structural or non-structural use. The ability of FRFC to have various densities and strength is making it readily applied in construction industry. The addition of fibres is altering the mechanical properties of FRFC while might give a minimum effect on functional properties, as the thermal and acoustic behaviours are relied on their pores quantity and its discontinuity. The empirical predictions of elastic modulus, compressive, flexural and tensile strengths for LWFC are found inaccurate for FRFC. Further investigations should be conducted for design application. Basically, the fibres inclusion may enhance the strength properties and durability of FRFC, while synthetic fibres may exhibit higher strength as compared to natural fibres inclusion. 

Furthermore, for time dependency properties, fibres also improved its properties, such as drying shrinkage and creep. As the pores within concrete have non-bearing characteristic, they may experience more shrinkage and creep than normal concrete. To reduce this effect in loading stage, fibres may benefit on these behaviors. 

## 6. Applications of FRFC

Due to a lack of knowledge on the practical use and manufacturing techniques of many experts and the difficulty of obtaining structural strength, in the last 60 years, FC has been widely ignored for use in concrete structural applications (Figure 8) [218].

In most cases, FC was employed to fill voids and was used as thermal isolation and behaviour as a sound damper [7,218]. Developments in mechanical and chemical foaming methods, admixtures of cement composite, and other additives considerably amended FC’s constancy and hardened properties [75,219]. Simultaneously, the appropriate use of FC for structural concrete utilizations is well-known, and many investigation works have been concentrated on enhancing its mechanical properties [6,7,10,14,15,55,75,218,219,220]. However, FC’s application has been noticeable as widespread applications internationally, particularly in distress from housing scarcities or endangered to adverse climate, storms and tremors [42]. FC could be made with dry densities of 300–1850 kg/m³, at compressive strength of about 1 and 58 MPa, respectively [8]. FC has a superior function at resisting a fire. Its acoustical and thermal insulation characteristic marks it perfect for an extensive range of applications, from void filling to insulating roofs and floors. It is also mainly valuable for ditch restoration [7]. In addition to many other typical uses of FC are applied under concrete pavement, to avoid ice lurch in road and rail network, to protect narrow foundation structures and placements, to avert ice lurch underneath pile caps and ice jacking of thin piles, to apply as a mortar to seal unrestricted tubes and as backfill under suppressed oil ground components, to reduce the heat under warm oil containers and the reservoir seats and block cavities under floors and decrease the thermal gradient and thermal stress in warm concrete pits and consequently protect shallow [218]. In Arab countries, FC beneficial characteristics such as lightweight nature and thermal insulation completed it as an appropriate material applied to decrease the negative influence of earthquakes and resolve the adverse consequence of the changes on temperature [8]. Further, FC applications are economical when repair, retrofit, strengthening and rehabilitation of concrete structures [221]. Table 9 shows the application of FC.

As FRFC have a wide range of density and strength, it has potential to be used in various applications. Serving as a building material, FRFC may be used in structural primary members, such as beams, columns, and frame systems. In addition, FRFC can also be used in non-bearing applications of brick structures. There is a product, GRP reinforced foam cement board, developed by a Chinese company, for application to the wall surface with characteristics of low thermal conductivity and acoustic transmission. There are several sample houses that were developed using FRFC in countries like Malaysia, India, Sri Lanka, Indonesia, and Philippines, for which the information are available through open online resources. 

## 7. Future Research Focus

Kum [222] found that shrinkage cracking, which developed in the FC, may become an issue in the future of material research. Creep strain is another parameter that is rarely found in current research trends. A gradual increment of strain for concrete in a function of time is referred to as a creep deformation. Long-term behaviour needs to be investigated as it is affecting the overall structural performance of FC members. Creep and shrinkage prediction models should be developed for different structural concrete member design. As the concrete matrix of lightweight aggregate concrete varies with FC, the interaction of steel reinforcements and a concrete matrix may also exhibit different structural performance. The pores minimize this interaction, which may result in slipping between these two materials. 

For shear behaviour of a reinforced beam, the design remains in doubt for structural members. For the deflection check and crack control, there remains uncertainty for further investigations. As found in previous research, the cracks were found more than in normal-weight concrete at ultimate strength, leading to excessive deformation of reinforced FC structures. Confinement with steel strap or other materials may mechanically increase the member capacity while reducing cracks at the serviceability state. 

The large volume of voids may promote electrochemical movement and prevent the passive layer [223]. Therefore, corrosion may take place and reduce the strength of RFC. The research direction of this passive layer is essential in assisting designers for reliable foamed concrete building construction. A life-cycle-assessment has been done for lightweight aggregate concrete blocks in China’s production [224]. From the life cycle stages of raw material achievement, production, use/recycle/repairs, and reutilize/waste management [224,225], lightweight concrete seems to require less energy, which leads to more sustainable practice. Another research also highlighted on the environmental impacts of FC production [226]. Reducing carbon footprint was discovered and more sustainable than autoclaved aerated concrete for wall construction [226]. Therefore, it is essential to identify the lightweight FC’s sustainability through a life-cycle-assessment, as it is claimed as a green constructional material.

## 8. Conclusions

The properties of FC, which make it attractive for potential use in construction, are briefly summarized. It has been shown that the addition of fibres of various origins have a high beneficial effect in enhancing the mechanical characteristics of FC by optimizing the microstructure of FC mass to allow its application for use in load-bearing structures. Natural and artificial fibres that are capable of increasing the efficiency of FRFC along this path have been classified. The factors influencing the properties of FRFC have been revealed and discussed, the most important of which is the density in the fresh and hardened state; type, diameter, content and length of fibre; the size of the aggregates; and the percentage of pozzolanic supplements. By varying these characteristics, optimal fresh state properties (consistency, rheology, workability, segregation and water separation) and hardened state properties (compressive, tensile, bending strength, impact viscosity, and modulus of elasticity, water absorption, shrinkage and thermal conductivity) can be achieved. Based on this study, it is found that the inclusion of fibres significantly enhanced the strength properties and durability of FRFC, in particular, splitting tensile and flexural strengths. It is also exhibited that the synthetic fibres revealed higher strength as compared to the addition of natural fibres. Furthermore, for time dependency properties, fibres also improved its properties, such as drying shrinkage and creep. As the pores within concrete have non-bearing characteristic, they may experience more shrinkage and creep than normal concrete. By systematically enhancing a number of these properties, it is found that there is a potential use of FRFC to produce RC applications to effectively accomplish both structural efficiency and thermal performance. In addition, extensive reviews of literature to furnish the potential application of FRFC composites in the building industries were widely presented. It can be concluded, however, that FRFC is nowadays a better construction material for field applications in the construction industries. This work also offers several recommendations for future studies,
-Future investigation to study the impact strength of FRFC.-Extra efforts must be paid to investigate the thermal insulating characteristic of FRFC that could be made by using the mixed-foaming techniques. Such foams have no superior characteristics than those of conventional porous materials, including glass foam or autoclaved-aerated concrete.-Further investigations to explore the possibility of utilizing FRFC in the construction of eco-friendly buildings, as applicable material to mitigate noise pollution for increase productivity, health and well-being.

## Figures and Tables

**Figure 1 materials-13-04323-f001:**
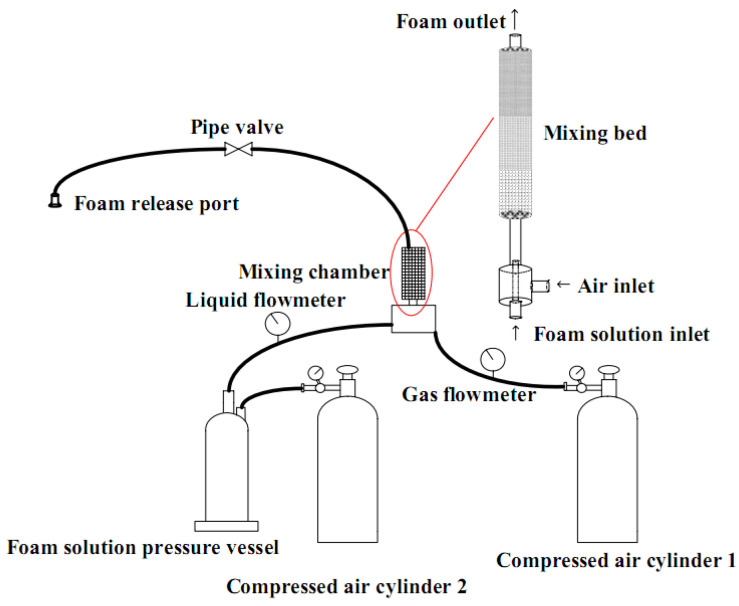
Schematic diagram of the compressed-air foam generator [73]. Reprinted with permission from Xiujuan et al. (2020) [73].

**Figure 2 materials-13-04323-f002:**
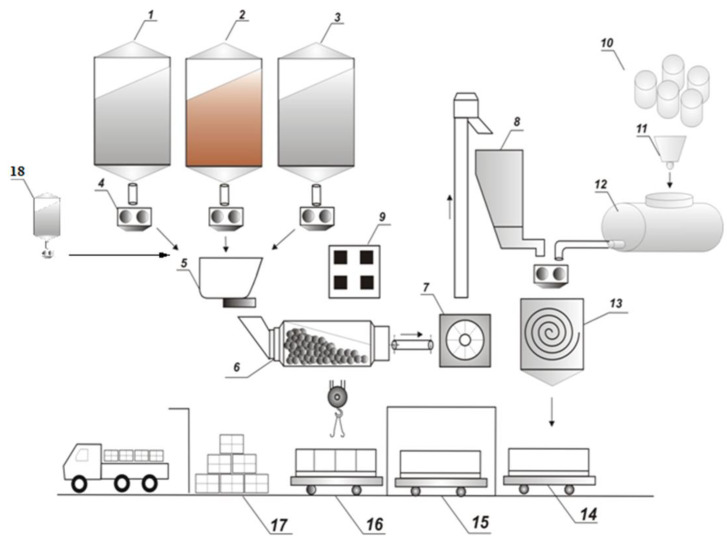
Technological route of FC manufacturing: **1**—PC bin; **2**—marl bin; **3**—binder bin; **4**—weighers; **5**—mill bin; **6**—mill; **7**—air pump; **8**—blended binder bin; **9**—control room; **10**—foamer; **11**—volumetric distributor; **12**—foam producer; **13**—FC blender; **14**—moulding zone; **15**—strength gain zone; **16**—stripping zone; **17**—packing and storing; **18**—superplasticizer bin [5]. Reprinted with permission from publisher (2020) [5].

**Figure 3 materials-13-04323-f003:**
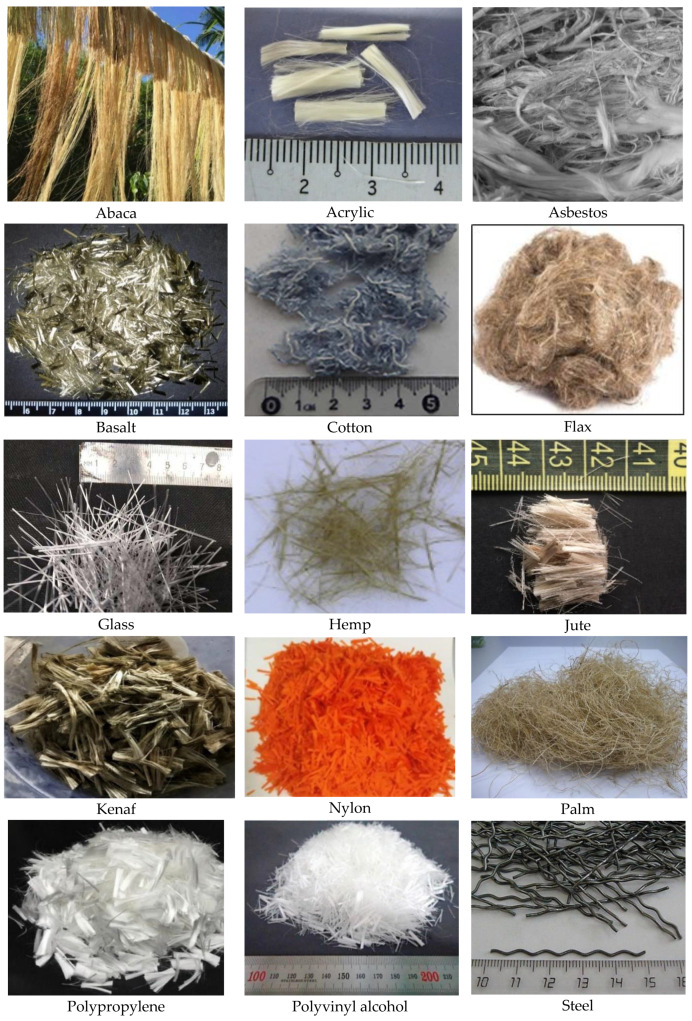
The appearance of some types of fibres.

**Figure 4 materials-13-04323-f004:**
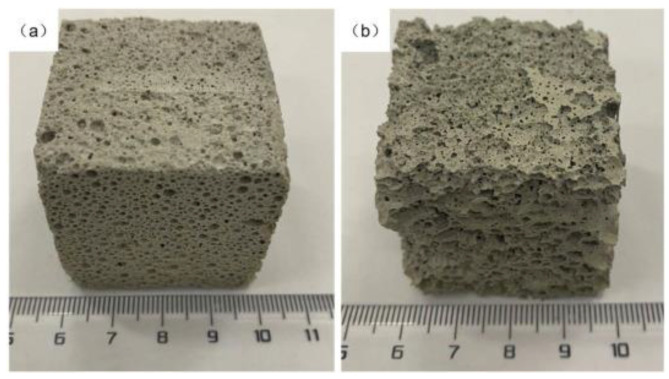
Air bubbles distributed in harden foamed concrete; with (**a**) ternary and (**b**) binary or without aerogels [168]. Reprinted with permission from publisher (2020) [168].

**Figure 5 materials-13-04323-f005:**
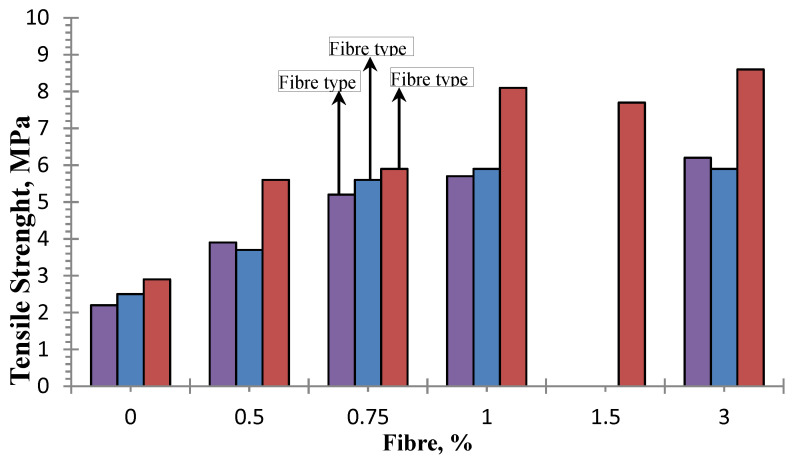
Tensile strength versus type of PP fibers [18]. Reprinted with permission from publisher (2020) [18].

**Figure 6 materials-13-04323-f006:**
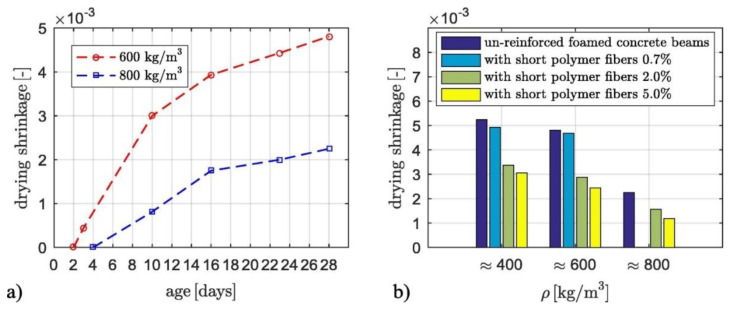
Evaluation of FC drying shrinkage (**a**) without fibres and (**b**) with fibres. Reprinted with permission from publisher (2020) [23].

**Figure 7 materials-13-04323-f007:**
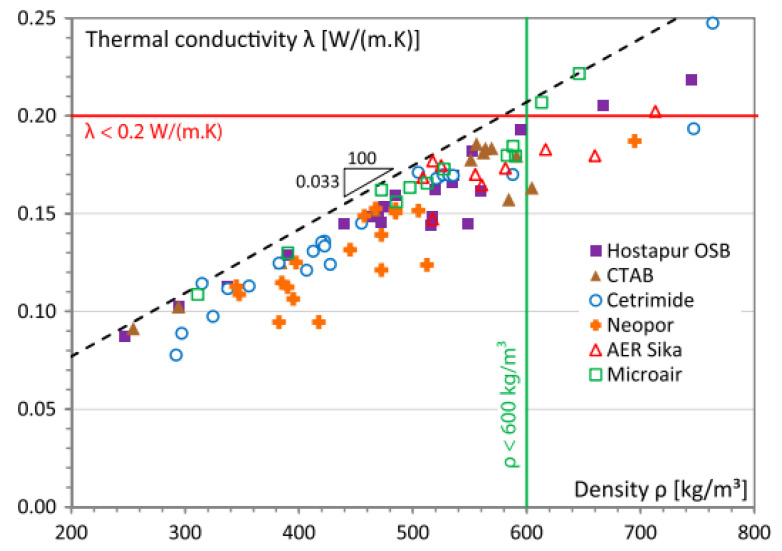
FC thermal conductivity versus its foam density [206]. Reprinted with permission from Samson el al. (2017) [206].

**Figure 8 materials-13-04323-f008:**
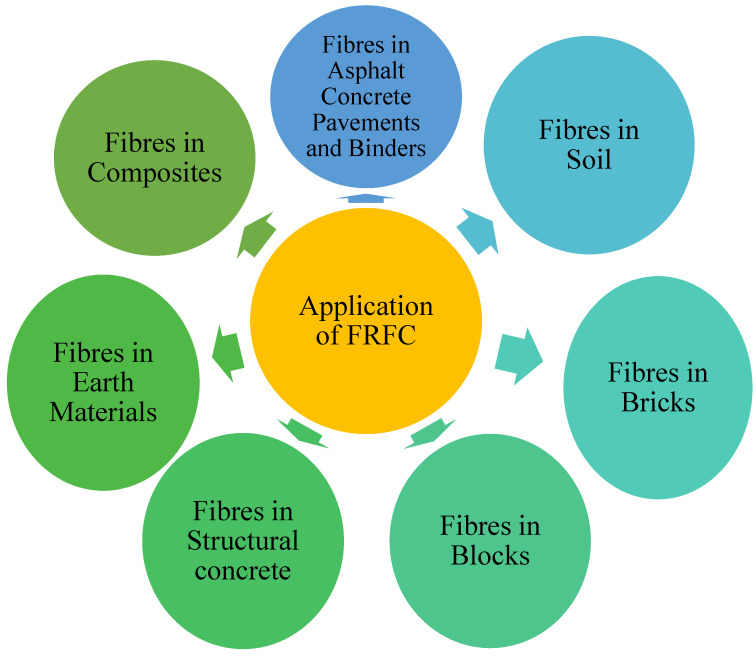
Application of FRFC.

**Table 1 materials-13-04323-t001:** Various foam agents are available in E-markets worldwide.

Name of Foam Agent	Property		Advantages	Density of		Online Access
	Natural	Synthetic		Foam	Concrete	
**Genfil** Herbal resin based	√	-	Improved high-yield herbal resin based foam agent Stable foam	80 to 95 g/L	115–1600 kg/m^3^	http://www.foam-concrete.com
**LithoFoam** protein based	√	-	Improved silicone oil resistance, frost resistance -Highly active proteins	20–180 kg/m³	1600–1675 kg/m³	http://www.luca-industries.com
**CMX™** Synthetic based	-	√	Performs well with a wide variety of ad-mixtures Proven to withstand higher lifts	1.02 kg/L	500–1600 kg/m³	https://www.richway.com
**Sakshi CLC** Synthetic Based	-	√	Air entrainers Set accelerator Water Reducer	0.2–0.7 L/m³	300–1300 kg/m³	https://www.sakshichemsciences.in/
**EABASSOC** Synthetic based	-	√	It highly concentrated, highly efficient liquid	0.3–0.6 L/m³	250–1800 kg/m³	https://www.eabassoc.co.uk
**VariMax** Synthetic based	-	√	To offer a variable high dilution ratio	1:40	150–1450 kg/m³	https://www.vermillionassociates.com
**LITEBUILT** Synthetic based	-	√	Fast turnaround in the production process No harmful or toxic fumes	2–3 wt.% of the mixture	300–1600 kg/m^3^	http://www.litebuilt.com/

**Table 2 materials-13-04323-t002:** Properties of FC with different dilution of foaming agent.

Refs.	Density kg/m^3^	Volume of Foam Agent		Type Materials Added	Compressive Strength at 28 Days
		By kg/m^3^	By Dilution with Water		
[63]	600	75–80 g/L	1:33	Lightweight aggregate, PP fibres, sand, and cement	25–58
	541–1003	-	0.5–3%	Sludge aggregate	25
[64]	1000			Cement–sand	1.82–16.73
[65]	1150	75–80 g/L	-	Fly ash, sand and cement	10–26
[55]	982–1185	40	0.5–3%	Fly ash sand, and cement	1.0–6.0
[66]	650–1200	40	1:5	Fly ash, sand, cement	20–43
[55]	280–1200	40	1:5	Fly ash, sand and OPC	0.6–10_91days_
[20]	800–1350	40	1:5	Silica fume (10–15%)	P4.73
	1380	0.25%		Fine sand, fly ash, lime, and PP fibre	15–30_77days_ 0.2–1_180days_ 1.6–4.6_180days_
[66]	800–1500	70	1:40	PP fibres, sand and cement	10–50
[54]		70	1:40	Course sand and OPC	1.0–7.0
[19]	650–1200	40	1:5	Partially (OPC-fly ash)	2.0–18
[29,56]	1000–1500	70	-	Fly ash, ultra-fine silica powder, and silica fume	85.4_365days_
[36]		70	1:5	Fly ash (fine and coarse)	4.0–7.3_7days_ 1.0–2.0_7days_ 0.5–10_7days_
[55,56]				PP fibre and Silica fume	39.6–91.3
[67]	1000–1400	50	-	Fly ash, cement and sand	4.0–19
[68]	1400	70	1:2		5.5–9.3
	1200–1600	70	-	Fine sand and OPC	2.0–11
	1710	50	-	Fly ash, fine sand, and	5.4–13.2
	400–1800	50	-	Fly ash, sand, and cement,	44_180days_
[19,29]	1400–1800	50	1:35	Lightweight aggregate, sand, and cement	9.9–39.5
[10]		59			13.8–48
[55]		50	-	Fly ash only	25
[69,70]		80	1:35	75%-fly ash, sand, and cement	40
[71]	1500–1800	60	-	Sand, aggregate and cement	1.8–17.9
[72]	1837	30–50	1:5		28

All reviewed compressive strength were tested at 28 days otherwise, it is mentioned beside the values.

**Table 3 materials-13-04323-t003:** Comparison on foam extrusion, batch foaming, and foam injection moulding [74].

Criterion	Foam Extrusion	Foam Injection Molding	Batch Foaming
Quantity of materials	Large, in kilograms	Larger amount, in kilograms	Less, in grams
Pre-molding	Not required	Not required	Required
Specimen state during gas saturation/loading temperature	Melt	Melt	Solid
Range of cell density (cells/cm^3^)	1 × 10^4^ to 1 × 10^11^	1 × 10^4^ to 1 × 10^8^	1 × 10^6^ to 1 × 10^16^
Distribution of cell	Uniform, but the cells in the core may differ in size than the cells at edges	Difficult to produce foam with uniform cell	Uniform
Quality of the surface	Good and glossy	Poor	Good
Skin layer thickness (μm)	Thin	Thick	Thin
Incorporation of nucleation agent	Composition can be changed any time Can be introduced any time during processing	Can be introduced any time during processing	Foam composition is fixed from the very beginning
Supply of blowing agent	Foamer is metered but not more than the melt can take	Gusting agent is measured but not greater than the dissolve can engross under special treatment mode	Sample is saturated with foam agent until equilibrium is reached
Cost of tooling	Costly, depends on the capacity of machine	Costly, depends on the capacity of machine and the cost of mold	Cheap

**Table 4 materials-13-04323-t004:** Fibre properties from previous FRFC research.

Ref	Fibres	Fibre Properties					Properties of FRFC at Max. Compression Strength			
		Length	Density	Tensile	Elastic Modulus	Dia	Comp	Density	Tensile	Flexural
		mm	kg/m^3^	MPa	GPa	µm	MPa	kg/m^3^	MPa	MPa
[1]	Polypropylene	19	900	552	3.8	70	2.14	734	0.341	-
	Henequen	19	1400	500	13.2	170	1.78	728	0.445	-
[23]	Polymer	20	1000	520		0.54	12.44	835	-	2.35
	Glass mesh	4 × 4 mm	125 g/m^2^	25	-	-	-	-	-	-
	Glass mesh + polymer	-	-	-	-		11.67	822	-	7.05
[80]	Polypropylene	12	1000	2.69	3.46	587	-	-	-	-
	Polyvinyl alcohol	25	1500	0.9	29	398	-	-	-	-
	Polypropylene	12	15	2.79	3.48	25	-	-	-	-
[72]	Polyolefin	50.4	-	275	2.6	0.64	7.82	1600	-	1.53
[29]	Polypropylene	15	900	800	8.0	100	50	1500	8.0	-

**Table 5 materials-13-04323-t005:** Hardened and physical properties of some fibres used in FC.

Sort of Fibre	Physical Properties		Hardened Properties		Refs.
Sisal fibre	Diameter	0.004–0.3 mm	Modulus of elasticity	5–18 GPa	[106]
	Length	4–160 cm			
	Absorption	110%	Tensile strength	233–580 MPa	
	Absolute density	1370 kg/m^3^			
Date palm fibre	Diameter	0.1–0.8 mm	Elastic modulus	5 GPa	
	Length	2.5–3.5 cm	Split strength	233 MPa	
	Apparent density	512–1089 kg/m^3^	-	-	
	Absorption	97–203%	-	-	
	Absolute density	1300–1350 kg/m^3^	-	-	
Date palm fibre	Diameter	60 mm	Split strength	240 ± 30 MPa	[107]
		20 mm		290 ± 20 MPa	
		100 mm		170 ± 40 MPa	
	Absolute specific weight	1300–1450 kg/m^3^	-	-	
	Apparent specific weight	512.21–1088.81 kg/m^3^	Elongation at break	0.23%	
	Humidity	9.5–10.5%	-	-	
	Water absorption (1 day)	96.83–202.64%	-	-	
Jute fibre	Diameter	1 mm	Elastic modulus	10–30 GPa	[106]
	Length	2–4 cm	Split strength	400–800 MPa	
	Absolute density	1700 kg/m^3^	-	-	
Hemp hurds	Diameter	1–8 mm	-	-	
	Length	0.5–3.5 cm			
	Absorption	280%			
Flax fibre	Diameter	0.035 mm	Elastic modulus	21 GPa	
	Length	7–8.5 cm	Split strength	805 MPa	
Kenaf fibre	Diameter	0.13 mm	Elastic modulus	136 GPa	
	Length	3 cm	Tensile strength	1000 MPa	
	Absorption	307%	-	-	
	Absolute density	1040 kg/m	-	-	
	Apparent density	50–114.4 kg/m^3^	-	-	
	Diameter	0.025–0.05 mm			
	Absolute density	440 kg/m^3^			
	Thermal conductivity	35.3–53.9 W/mK			
	Absorption	240%			
Polypropylene and polyethylene fibres	Aspect ratio	90	-	620 MPa	[108]
	Length	40 mm	Split strength	600 MPa	
	Specific gravity	0.92	Elastic modulus	9.5 GPa	
	Thickness	0.105 mm	-	-	
	Cross-section shape	Rectangular	-	-	
	Width	1.4 mm	-	-	
PET fibre	Density	1.38 g/cm^3^	Split strength	3 GPa	[109]
				10 GPa	
				160 MPa	
				420 MPa	
				450 MPa	
			Elastic modulus	-	
Polyester fibre	Diameter	30–40 µm	Elastic modulus	10–30 GPa	[110]
	Length	3, 6, 12, 20, 64 mm	Split strength	400–600 MPa	
	Specific gravity	1.35 g/cm^3^	-	-	
Steel fibre	Diameter	0.6 mm	Split strength	>1000 MPa	[111]
	Length	32 mm			
	Specific gravity	7.85			
Steel fibre	Maximum diameter	0.75% ± 5% mm	Elastic modulus	200,000 MPa	[112]
	Length	60% ± 5% mm	Tensile strength	1035 MPa	
	Number of fibres per kg	4600	-	-	
	Slenderness ratio	90	-	-	
Glass fibre	Diameter:	3–19 µm	Elastic modulus	53–95 GPa	[110]
	Length	25 mm	Ultimate tensile strength	1500–5000 MPa	
	Specific gravity	2.49–2.60 g/cm^3^	-	-	
E-glass fibre	Density	2.61 g/cm^3^	Modulus of elasticity	57.0 ± 3.0 GPa	[113]
	Diameter	16.8 ± 1.6 µm	Split strength	1472 ± 395 MPa	
	Area or cross-section	223.4 ± 42 µm^2^	Maximum force	0.32 ± 0.08 N	
Asbestos	Specific surface area	60 (10^−3^ m^2^/g)	Split strength	602 ± 295 MPa	[114]
	Average length	5.5 mm			
	Specific gravity	2.75			
Basalt fibre	Density	2.66 g/cm^3^	Split strength	30–40 MPa	[113]
	Diameter	10 ± 3.1 µm	Elastic modulus	48.2 ± 20.6 GPa	
	Cross-section	90.2 ± 56.7 µm^2^	Maximum force	0.05 ± 0.04 N	
Propylene nylon fibres	Thickness	80 to 1500 µm	Specific tensile modulus	0.27–0.7 GPa/g/cm^3^	[115]
	Length	2 to 20 mm			
	Water absorption	0–4.5%			
	Specific gravity	0.9–1.32			

**Table 6 materials-13-04323-t006:** Properties of different FRFC composites and mortars.

Refs.	Mix Details			Fresh Properties	Harden Properties					
					Mechanical					Functional
	Fibre	Design	Optimum	Density kg/m³	Density kg/m³	Compressive Strength MPa	Tensile Strength MPa	Flexural Strength MPa	Shrinkage	Thermal Conductivity
[121,122]	paper fibres	cement: sand: w/c ratio 1:1.5:0.45 (5, 10, 15, 20% of paper fibres)	0.2	1043–1099	930–1060	0.5–1.9	-	0.15–0.8	-	-
[32]	MC-8B waste paper (recycled cellulose fibres)	0.1–0.5% of binder mass	0.003	-	D600	1.9 (0.3%, 28 d)	-	-	decrease	-
[123,124]	Basalt	-	0.001	-	D400	60% increment to CS (0.1%, 28 d)	-	-	57% reduce	35% reduce
	Chrysotile asbestos	-	0.02	-	D400	2.5 times to CS (2.0%, 28 d)	-	-	90% reduce	40% reduce
	Polypropylene	-	0.004	-	D400	62% increment to CS (0.4%, 28 d)	-	-	42% reduce	30% reduce
[72]	Polyolefin	0.2, 0.4, 0.6, 0.8%	0.004	-	1600	6.397–7.8236	-	1.36–1.53	-	-
[23,38]	short polymer fibres and GFRP	0.7, 2.0, 5.0%	2.0% polymer fibres + GFRP	400, 600, 800	382–823	1.11–12.84	-	0.09–2.53	-	-
[35]	carbon fibres and polypropylene fibres	0.5, 1.0, 1.5%	1% of carbon + 0.5 polypropylene (cost effective)	1800	1670–1820	13.8–23.1	1.9–2.9	2.1–4.5	-	-
[125]	coir and polyvinyl alcohol fibres	0.3, 0.4, 0.5%	0.3% of coir + 0.2% of PVA or 0.2% of coir + 0.3% of PVA	-	-	6.5–16	2.0–4.5	1.0–2.4	-	-
[126,127]	Basalt, Polyvinyl alcohol (PVA), polypropylene (PP)	2%, alkali-activated slag foam concretes	PVA	-	-	32–40	-	2.5–8	-	-
[128,129,130]	Polyolefin	<0.55%	hybrid (macro + fibrillated)	-	-	3.89–8.44	-	6.3–10.68	-	-
[131]	steel fibres	0.5–1.5%	-	2150–2250	-	17.34–22.37	0.88–2.97	-	-	-
[27]	kenaf and polypropylene	0.25 & 0.4%	-	1130	-	2–3.8	-	0.55–1.0	0.1–0.4 mm	-
[132]	Polypropylene	0.2, 0.25, 0.3%	0.002	1600	-	7.459–10.783	-	-	-	0.66–0.71 W/mK
[80]	Polypropylene	0.5–3%	0.008	1.0001 × 10^15^	-	25,842	2.6–4.0	41,306	55% reduce	0.56–0.64 W/mK
[1]	Henequen	0.5, 1, 1.5%	-	700	681-732	1.42–2.14	0.225–0.447	-	-	-
[29]	Polypropylene	-	0.008	800–1500	-	18,537	3.5–8.0	-	1.3–1.7 mm	-

**Table 7 materials-13-04323-t007:** Effect of the grading of aggregates.

Type of Material Used	Main Findings	Refs.
Quarry waste	The excellent bond is achieved through the finer quarry dust, which alleviated the necessity of foam’s volume for the given density of concrete. Henceforth, improved compressive strength and thermal conductivity were detected.	[136]
Fine-recycled concrete aggregate	Enhanced mechanical properties at 10% replacement by sand weight. Simultaneously, recycled sand demonstrates more excellent water absorption and porosity in compared to calcareous sand used.	[137]
Polyvinyl waste	The combined content of Fe_2_O_2_, Al_2_O_3_ and SiO_2_ in polyvinyl waste above 50% promotes creating a C-S-H gel, resulting in improved characteristics of both flexural and compressive strength.	[138]
Rice husk ash (RHA)	Mechanical properties improve with increasing content of RHA up to 20 wt.% due to its pozzolanic nature.	[22]
M Sand	Mechanical properties improve with increasing content of M Sand up to 20 wt.%.	[139]
Biomass aggregates	The biomass aggregate in FC results in superior compressive strength when exposed to air for 91 days compared to conventional fine aggregate.	[140]
Different gradations of sand	Mechanical properties enhance with the fineness of aggregate.	[141]
River sand, sea sand and quarry dust	The sample with quarry dust as a fine aggregate achieved higher strength and density than the other samples, and sea sand as a filler reached very similar strength and density values with river sand.	[142]
Glass fines	The shrinkage of the cement paste is reduced with a further improvement in strength at low density.	[143]

**Table 8 materials-13-04323-t008:** Prediction models of different FRFCs.

Ref.	Elastic Modulus–Prediction Model		
	[21,192]	[192]	[63]
	$E=33W^{1.5}f_{c}{}^{0.5}$	$E=0.99f_{c}{}^{0.67}$	$E=6326\rho_{con}^{1.5}f_{c}{}^{0.5}$
	GPa	GPa	GPa
[121,122]	0.7	0.6	4.0
[32]	0.7	1.5	4.1
[72]	5.3	3.4	32.4
[23,38]	0.3	1.1	1.6
[35]	8.4	5.7	50.7
[125]	-	3.5	-
[126,127]	-	10.1	-
[128,129,130]	-	2.5	-
[131]	13.7	6.7	83.0
[27]	1.8	1.6	10.7

Annotations: *w* = unit weight of concrete, Kg/m^3^, *f_c_* = compressive strength of concrete, MPa, and *ρ* = density, kg/m^3^.

**Table 9 materials-13-04323-t009:** Applications of FC.

Refs.	Density, kg/m^3^	Applications
[8]	<300	Production of decoration applications and partitioning walls
[75]	300–600	Substitutions of current soil, raft foundation, soil steadiness.
[8]	500–600	Rehabilitation of geotechnical applications including road construction and soil settlement,
[219]	600–800	Void filling, including old sewerage pipes, boreholes, basement and tunnels
[55]	800–900	Fabrications of masonry blocks, bricks for non-load bearing walls in building constructions
[10]	1100–1400	Fabrications of non-and-load bearing applications including floor screeds
	1100–1500	Housing construction applications
[219]	1600–1800	Production of wall bearing and slabs systems of concrete building
[6,7,14,15,220]	1800–1900	Production of load-bearing precast sandwich wall panels

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
