# Peer review of "Fibre-Reinforced Foamed Concretes: A Review"

_materials, 2020, doi:10.3390/ma13194323_

Round 1
Reviewer 1 Report
This study collected and summarized the use of fiber-reinforcement for foamed concrete. The main composition looks fine and can be used to contribute to the further development of advanced foamed concrete. Nevertheless, some content should be improved critically to be published in Materials journal. Some comments on the manuscript as follows: 1. First, the reviewer recommends the authors to take an English review from a native speaker since some unnatural expressions and grammar can be found in the manuscript. It is strongly recommended to review the manuscript by a professional proofreading service. 2. Abstract: the abstract is too long and not reader-friendly. It seems like the authors have just listed the general use of foamed concrete with fiber-reinforcement. 3. Some formats look totally careless. For example, the fonts size and the format of Tables 4 and 7 are totally different. The authors need to unify such content. In addition, please unify the fonts and the general features of all figures. 4. In the title, the authors mentioned that this study is of a review of ‘fibre-reinforced foamed concrete’. However, many citations are of just foamed concrete without the use of fiber for reinforcement purposes. The authors should clarify this point in the introduction as well as the whole manuscript. 5. Though it is a review paper, there is no clue of a detailed discussion about the previous studies. Not only the results presented in the references, but also the authors need to include additional discussion, at least Sections 4, 5, and 6. 6. The format of the references should be corrected. For instance, the author lists were presented with their full name, while some of the others were in abbreviation. It is the same for journal titles as well.Author Response
|
Reviewer # 1 |
||
|
Comment No. |
Reviewer 1 Comments |
Response to Reviewer 1 |
|
|
|
|
|
2. |
Abstract: the abstract is too long and not reader-friendly. It seems like the authors have just listed the general use of foamed concrete with fiber-reinforcement. |
The abstract has been completely rewritten to provide the reader with the most complete picture of the article. |
|
3. |
Some formats look totally careless. For example, the fonts size and the format of Tables 4 and 7 are totally different. The authors need to unify such content. In addition, please unify the fonts and the general features of all figures. |
The problems with fonts and formatting of all Tables (1-9) and Figures have been corrected. |
|
4. |
In the title, the authors mentioned that this study is of a review of ‘fibre-reinforced foamed concrete’. However, many citations are of just foamed concrete without the use of fiber for reinforcement purposes. The authors should clarify this point in the introduction as well as the whole manuscript. |
Accordingly, the required clarification and the needful information have been successfully added to the manuscript.
For the sake of clarification, this review study exhibited that there were only limited number of studies on FRFC as well as to introduce the production techniques of foaming agent and FC technology, thus, this leads the review of the literature to also include conventional FC to be able to draw a more complete picture of the current state-of-the-art. |
|
5. |
Though it is a review paper, there is no clue of a detailed discussion about the previous studies. Not only the results presented in the references, but also the authors need to include additional discussion, at least Sections 4, 5, and 6. |
Discussions have been added to Sections 4, 5 and 6. A brief summary has also been included in these sections. |
|
6. |
The format of the references should be corrected. For instance, the author lists were presented with their full name, while some of the others were in abbreviation. It is the same for journal titles as well. |
The style of the references of ‘Materials, MDPI’ was automatically formatted using Mendeley software, as shown below |

Reviewer 2 Report
Author recommend the following corrections for this manuscript
- Line 401 please specify the missing reference.
- Line 413 Please modify the table 7.
- Line 423 please specify the missing reference.
- Line 435 After table 7, table 6 is mentioned, please correct it
- Line 611 please specify the missing reference.
- Line 614 please modify the table 8
- In figure 6 Please mention the units of drying shrinkage (if any) in Y-axis.
- Figure8 should be moved to the introduction part.
Author Response
|
Reviewer # 2 |
||
|
Comment No. |
Reviewer 2 Comments |
Response to Reviewer 2 |
|
1. |
Line 401 please specify the missing reference. |
The missing was a citation for ‘Table 6’ in the text and it has been corrected. |
|
2. |
Line 413 Please modify the table 7. |
Table 7 has been modified as per the reviewer’s suggestion. Table 7 has been renumbered as Table 6 in the revised manuscript.
|
|
3. |
Line 423 please specify the missing reference. |
The missing was a citation for ‘Table 3’ in the text and it has been successfully corrected. |
|
4. |
Line 435 After table 7, table 6 is mentioned, please correct it |
Table numbers 6 and 7 have been corrected as per the reviewer’s suggestion. |
|
5. |
Line 611 please specify the missing reference. |
The missing reference has been added. |
|
6. |
Line 614 please modify the table 8 |
Table 8 has been modified as per the reviewer’s suggestion. |
|
7. |
In figure 6 Please mention the units of drying shrinkage (if any) in Y-axis. |
The shrinkage values were given as strains, and hence they had no units. Only given as “µ” that has been added to the figure. |
|
8. |
Figure8 should be moved to the introduction part. |
The application of FRFC was discussed in section 6. Figure 8 describes an application of FRFC, and hence it is more appropriate to present this figure under this section. |

Reviewer 3 Report
The review paper discussed different angles of fibre-reinforced foam concrete. The work needs the following revision before publishing.
- The English language of the manuscript requires an extensive revision by professionals. There are many structural and grammatical mistakes within the text, which creates ambiguity.
- Line 18-20: The meaning is not clear. Please revise and clarify the sentence.
- Line 139: Grammatical mistake
- Line 149: Grammatical mistake
- Line 174: Change 2.2. Methods of foaming generation to 2.2. Methods of manufacturing foam
- Line 179: fly ash is uncountable
Author Response
|
Reviewer # 3 |
||
|
Comment No. |
Reviewer 3 Comments |
Response to Reviewer 3 |
|
1. |
The English language of the manuscript requires an extensive revision by professionals. There are many structural and grammatical mistakes within the text, which creates ambiguity. |
The English language has been improved through a language editing service. |
|
2. |
· Line 18-20: The meaning is not clear. Please revise and clarify the sentence. |
In line 18-20 has been rewritten clearly as follows:
FC comprises no hazardous substances that might be environmentally unfriendly, making it an eco-friendly material for building applications. Since FC may consist of up to 80 percent air, it has excellent fire-resistance, and sound-absorption and insulation properties. However, the utilization of FC as structural concrete has been limited due to its low tensile strength and the fact that in a plastic state it often cracks in the course of drying shrinkage.
|
|
3. |
· Line 139: Grammatical mistake |
In line 139, the grammatical mistake has been corrected as follows:
The quality of the foam agent significantly influence the strength and toughness of FC [56]. The volume of bubbles typically vary from 6% to 35% of the total final mixture in most of FC applications [57].
|
|
4. |
· Line 149: Grammatical mistake |
In line 149, the grammatical mistake has been corrected as follows:
Table 1. Various foam agents are available in E-markets worldwide |
|
5. |
· Line 174: Change 2.2. Methods of foaming generation to 2.2. Methods of manufacturing foam |
Line 174 has been changed as per the reviewer’s suggestion.
|
|
6. |
· Line 179: fly ash is uncountable |
The required correction has been made and the needed clarifications have been added. |

Round 2
Reviewer 2 Report
Can be accepted in the present form.